# Enolase Inhibitors as Early Lead Therapeutics against *Trypanosoma brucei*

**DOI:** 10.3390/pathogens12111290

**Published:** 2023-10-28

**Authors:** Colm P. Roster, Danielle LaVigne, Jillian E. Milanes, Emily Knight, Heidi D. Anderson, Sabrina Pizarro, Elijah M. Harding, Meredith T. Morris, Victoria C. Yan, Cong-Dat Pham, Florian Muller, Samuel Kwain, Kerrick C. Rees, Brian Dominy, Daniel C. Whitehead, Md Nasir Uddin, Steven W. Millward, James C. Morris

**Affiliations:** 1Eukaryotic Pathogens Innovation Center, Department of Genetics and Biochemistry, Clemson University, Clemson, SC 29634, USA; croster@g.clemson.edu (C.P.R.); dalavig@g.clemson.edu (D.L.); jmilane@g.clemson.edu (J.E.M.); ewknigh@g.clemson.edu (E.K.); heidid@g.clemson.edu (H.D.A.); ssutto3@clemson.edu (S.P.); hardinel@musc.edu (E.M.H.); mmorri3@clemson.edu (M.T.M.); 2Department of Cancer Systems Imaging, UT MD Anderson Cancer Center, Houston, TX 77030, USA; victoriacyanide@gmail.com (V.C.Y.); cpham3@mdanderson.org (C.-D.P.); mnuddin@mdanderson.org (M.N.U.); smillward@mdanderson.org (S.W.M.); 3Sporos Bioventures, 3000 Bissonnet, Belmont Suite 5303, Houston, TX 77005, USA; aettius@aol.com; 4Eukaryotic Pathogens Innovation Center, Department of Chemistry, Clemson University, Clemson, SC 29634, USA; skwain@g.clemson.edu (S.K.); krees@g.clemson.edu (K.C.R.); dwhiteh@clemson.edu (D.C.W.); 5Department of Chemistry, Clemson University, Clemson, SC 29634, USA; dominy@g.clemson.edu

**Keywords:** enolase, African trypanosome, glycolysis, inhibitors, *Trypanosoma brucei*

## Abstract

Glucose metabolism is critical for the African trypanosome, *Trypanosoma brucei*, serving as the lone source of ATP production for the bloodstream form (BSF) parasite in the glucose-rich environment of the host blood. Recently, phosphonate inhibitors of human enolase (ENO), the enzyme responsible for the interconversion of 2-phosphoglycerate (2-PG) to phosphoenolpyruvate (PEP) in glycolysis or PEP to 2-PG in gluconeogenesis, have been developed for the treatment of glioblastoma multiforme (GBM). Here, we have tested these agents against *T. brucei* ENO (*Tb*ENO) and found the compounds to be potent enzyme inhibitors and trypanocides. For example, (1-hydroxy-2-oxopyrrolidin-3-yl) phosphonic acid (deoxy-SF2312) was a potent enzyme inhibitor (IC_50_ value of 0.60 ± 0.23 µM), while a six-membered ring-bearing phosphonate, (1-hydroxy-2-oxopiperidin-3-yl) phosphonic acid (HEX), was less potent (IC_50_ value of 2.1 ± 1.1 µM). An analog with a larger seven-membered ring, (1-hydroxy-2-oxoazepan-3-yl) phosphonic acid (HEPTA), was not active. Molecular docking simulations revealed that deoxy-SF2312 and HEX had binding affinities of −6.8 and −7.5 kcal/mol, respectively, while the larger HEPTA did not bind as well, with a binding of affinity of −4.8 kcal/mol. None of these compounds were toxic to BSF parasites; however, modification of enzyme-active phosphonates through the addition of pivaloyloxymethyl (POM) groups improved activity against *T. brucei*, with POM-modified (1,5-dihydroxy-2-oxopyrrolidin-3-yl) phosphonic acid (POMSF) and POMHEX having EC_50_ values of 0.45 ± 0.10 and 0.61 ± 0.08 µM, respectively. These findings suggest that HEX is a promising lead against *T. brucei* and that further development of prodrug HEX analogs is warranted.

## 1. Introduction

The African trypanosome, *Trypanosoma brucei*, is the protozoan parasite responsible for African sleeping sickness in humans and nagana in livestock animals. The intensive searches for new therapeutics against the parasite have led to the development of several very promising drugs for human use, including fexinidazole and acoziborole. Given the prevalence and importance of the infection in key food animal species, and concerns about the development of resistance to the new front-line human anti-trypanosomal drugs should they be employed against animal disease, the search for additional agents is warranted to expand the arsenal of useful agents against the pathogen.

*T. brucei* is transmitted by the bite of an insect vector, the tsetse fly. The female flies feed on mammalian blood and in the process can transmit parasites into the host. Once established in the bloodstream, the long slender bloodstream form (BSF) of the parasite divides rapidly and eventually overcomes the blood–brain barrier. Infection in the brain leads to classical symptoms associated with the disease, including lethargy, confusion, and wasting. If left untreated, infection is invariably fatal.

The BSF of the parasite is dependent on glycolysis as its sole source of ATP generation. This has led us and others to pursue trypanosome-specific inhibitors of glycolytic enzymes, an effort that has met with some success. A series of *T. brucei* hexokinase inhibitors have been described that are preferentially toxic to parasites over mammalian cells [1,2] and, more recently, allosteric inhibitors of trypanosome phosphofructokinase have been described that are curative in a rodent disease model [3].

Enolase, the enzyme that interconverts 2-phosphoglycerate (2-PG) to phosphoenolpyruvate (PEP) in glycolysis and gluconeogenesis, has been shown to be essential in *T. brucei*, with RNAi leading rapidly to BSF cell death [4]. This cytosolic protein, a 429-residue enzyme, is found as a single gene in the parasite, and sequence analysis and biochemical characterization have confirmed that residues for substrate binding and metal coordination are conserved with ENOs from other systems [5,6].

ENO inhibitors have proven useful as leads for the treatment of other diseases. For example, Lin and colleagues developed a series of phosphonate inhibitors of human enolase 2 (*Hs*ENO2) with the goal being the identification of brain tumor-specific therapeutics [7]. Most human tissues predominantly express *Hs*ENO1; however, *Hs*ENO1-deleted glioblastoma lacks this protein as it is proximal to the tumor-suppressor locus 1p36 and is a passenger deletion that is lost coincident with the locus. In its place, the tumor cells rely on *Hs*ENO2, a minor contributor to ENO activity in most tissues. To exploit this difference, which has been recognized as a therapeutic vulnerability [8], *Hs*ENO2-specific inhibitors were developed using structure-guided drug design, resulting in leads that selectively killed *Hs*ENO1-deleted glioma cells. These compounds have promising pharmacodynamic and pharmacokinetic properties and cure intracranial orthotopic *ENO1*-deficient tumors in mice while being very well-tolerated in non-human primates [7,9].

Here, we have explored the action of *Hs*ENO2-specific phosphonate inhibitors against the *T. brucei* ENO (*Tb*ENO). We hypothesize that these inhibitors will be potent enzyme inhibitors with anti-trypanosomal activity. The agents investigated here are potent enzyme inhibitors, with selectivity for the enzyme that differs from other ENOs due to differences in the substrate binding pocket. Further, these agents have promising in vitro activity against the BSF parasite, suggesting the agents could be useful leads in the development of new treatments for *T. brucei* infections.

## 2. Materials and Methods

Chemicals and Reagents—β-nicotinamide adenine dinucleotide (NADH), adenosine diphosphate (ADP), and dimethyl sulfoxide (DMSO) were obtained from Fisher Scientific (Pittsburgh, PA, USA). Other reagents were purchased from MilliporeSigma (Burlington, MA, USA) unless otherwise noted. The phosphonate ENO inhibitors (1-hydroxy-2-oxopyrrolidin-3-yl) phosphonic acid (deoxy-SF2312), (1-hydroxy-2-oxopiperidin-3-yl) phosphonic acid (HEX), and (1-hydroxy-2-oxoazepan-3-yl) phosphonic acid (HEPTA) were synthesized as described [7,9].

Recombinant ENO and ENO Assays—Recombinant *Tb*ENO was expressed and purified using an approach modified from the purification of another trypanosome glycolytic enzyme [10]. Briefly, the full-length open reading frame was cloned from *Trypanosoma brucei brucei* (a 427 strain) genomic DNA into the bacterial expression vector pQE-30 (Qiagen, MD, USA). This plasmid was transformed into M15pREP cells and protein expression induced with 0.25 mM isopropyl-b-d-1-thiogalactopyranoside (IPTG) overnight at 37 °C. Protein was purified on Ni-NTA agarose (Qiagen) as described [1].

ENO assays were performed in triplicate using a coupled reaction to measure enzyme activity. In short, enzyme (~50 nM) in assay buffer (100 mM HEPES, pH 8.0, 3.3 mM MgSO_4_, 120 mM KCl, 1.75 mM ADP, 1 U of pyruvate kinase/lactate dehydrogenase, and 0.4 mM NADH) was added to black 96-well plates. Reactions were initiated by the addition of substrate (3.75 mM 2-PG) and the rate of reduction in fluorescence emission at A_460_ was measured after excitation at A_360_ to monitor NAD^+^ production. Assays were scored on a Biotek Synergy H1 microplate reader and kinetic analyses were performed with Prism 9.0 (GraphPad Software, San Diego, CA, USA) using the Michaelis–Menten model. To assess the impact of different pH on enzyme activity, the HEPES (pH 8.5) in the standard buffer was replaced with Good’s buffers 2-morpholinoethanesulfonic acid (MES, pH 6.5), HEPES (pH 7.0 to pH 8.5), or sodium borate (pH 9.0). For assays involving inhibitors, compounds (in DMSO) were incubated with the enzyme in assay buffer (pH 8.0) for 15 min prior to initiation of the reaction. All assays were performed in triplicate with vehicle control (DMSO) included.

Ligand preparation, optimization, and molecular docking—The chemical structures of deoxy-SF2312, HEX, POMHEX, and HEPTA were drawn with ChemOffice professional 19 suite (PerkinElmer, Waltham, MA, USA), and three-dimensional (3D) structures were generated with VeraChem Vconf (VeraChem LLC, Germantown, MD, USA).

Full (unconstrained) geometry optimization of the 3D chemical structures was carried out in the gas phase (no implicit solvent) using the Gaussian 09 suite with DFT calculation, employing the B3LYP/6-311G (d,p) level of theory for geometry optimization and transition state study [11,12,13,14]. The geometry optimization was carried out to ensure the proper arrangement of all bond lengths, bond angles, and torsional angles in space since the drawn chemical structures are not necessarily energetically favorable. After the energy refinement of the chemical structures, vibrational frequency calculations performed on the optimized conformers indicated that the stationary points corresponded to minima on the potential energy surface.

The 3D crystal structure of *Tb*ENO (PDB 2PTY) was retrieved from the RCSB protein data bank. The protein was then prepared for the docking analysis through first removing co-crystallized ligands, heteroatoms, and water molecules using Pymol Molecular Graphics 2.0 (Schrödinger LLC, New York, NY, USA). The optimized ligands and the protein were further prepared using AutoDock Tools (The Scripps Research Institute, La Jolla, CA, USA). Only polar hydrogens were added to the protein, and Kollman united-atom partial charges were assigned. The ligands were treated as all-atom entities by initially adding all hydrogens, and then partial atomic charges were calculated using the Gasteiger–Marsili method. The prepared protein and ligands were then converted into pdbqt formats. A grid box was prepared around the region of the active site of the protein. The size of the grid box was kept at 40, 42, and 40 Å for X, Y, and Z, respectively, with the center of the grid box maintained at 11.399, 29.291, and 37.716 Å for X, Y, and Z, respectively, within the binding site. The output of the grid box was saved as a text file, after which it was imported together with the pdbqt files of the protein and ligands into AutoDock Vina (version 1.1.2). The molecular docking studies were carried out in vacuo (no implicit solvent) in AutoDock Vina which uses a united-atom vina scoring function and Lamarckian genetic algorithm search method. The applied scoring function accommodates ligand flexibility by considering bond rotations, translations, and whole conformational changes within the rigid binding site of the receptor during sampling [15]. This approach aims to identify the optimal binding pose for each ligand–receptor complex. The binding affinities of ligands were measured in kcal/mol as a unit for a negative score. The binding conformation with the most negative binding affinity was taken as the best pose for the corresponding protein–ligand complex. Subsequently, the best binding pose of each complex was analyzed using Pymol and Discovery Studio (Dassault Systèmes, Waltham, MA, USA) to reveal the protein–ligand interactions.

Trypanosome growth, viability assays, and Western blotting—*T. brucei brucei* 90-13 BSF parasites were cultured in HMI-9 medium supplemented with 10% heat-inactivated FBS and 10% Serum Plus (Sigma-Aldrich, St. Louis, MO, USA) at 37 °C in 5% CO_2_, while *T. brucei brucei* PF 29-13 were cultured in SDM-79 supplemented with 10% heat-inactivated FBS [16]. For BSF viability assays, parasites (1 × 10^5^/mL) were seeded into 384-well black polystyrene plates in the presence of a compound or equivalently diluted carrier. CellTiter Blue reagent (Promega, Madison, WI, USA) was added after two days of culture, the plates incubated an additional 1 h, and fluorescence emission at A_585_ measured after excitation at A_546_. For PF viability assays, 1 × 10^6^ parasites in standard medium, or 5 × 10^6^ parasites in glucose-free medium, were introduced into black 96-well plates in standard culture medium in the presence of a compound or carrier, with CellTiter Blue reagent added after 48 h of culture. Plates were incubated for 1 h and fluorescence emission scored as described for BSF. Fluorescence values were used to calculate percent cell growth inhibition (based on controls) for each concentration of compound tested. Averages of the triplicates were calculated and fit to dose-response curves for the determination of EC_50_ values, using Prism 9.0 (GraphPad Software, San Diego, CA, USA).

Trypanosome ROS and Annexin assays

BSF parasites (5 × 10^6^ cells) were pelleted (800 × *g*, 10 min) then resuspended in PBS supplemented with 0.1% glucose (PBSg). Samples were treated with either ROS stain (Total Reactive Oxygen Species (ROS) Assay Kit 520 nm, Invitrogen) or DMSO and then incubated at 37 °C in 5% CO_2_ for 60 min. Samples were then treated with 125 μM H_2_O_2_, 50 μM POMHEX, 100 μM POMHEX, or equal volume DMSO, incubated for 30 min, and analyzed on a CytoFLEX flow cytometer (Beckman Coulter). Events (10,000) were recorded and analyzed using FCSexpress software. Median fluorescence intensity (MFI) was calculated, and Welch’s *t*-tests were performed (ns: not statistically significant, * *p* < 0.05, ** *p* < 0.01, *** *p* < 0.001). Each experiment was performed with biological triplicates in technical triplicates.

To monitor apoptotic-like responses, 5 × 10^6^ parasites were collected, resuspended in fresh media containing 125 μM H_2_O_2_, 5 μM POMHEX, 50 μM POMHEX, 100 μM POMHEX, or equal volume of DMSO, and incubated at 37 °C in 5% CO_2_ for 30 min. Cells were washed once in PBS, resuspended in binding buffer (10 mM HEPES/NaOH, pH 7.4, 140 mM NaCl, 2.5 mM CaCl_2_) with propidium iodide (Enzo Life Sciences) and incubated for 15 min, followed by cytometry. Each experiment was performed with technical triplicates of biological triplicates. Median fluorescence intensity (MFI) was calculated, and Welch’s *t*-tests were performed (ns: not statistically significant, * *p* < 0.05, ** *p* < 0.01, *** *p* < 0.001).

## 3. Results

### 3.1. The Phosphonohydroxamates Inhibit TbENO In Vitro

The *Tb*ENO gene (Tb927.10.2890) was cloned into the bacterial expression vector pQE30 and heterologously expressed in M15pREP *E. coli.* Protein was purified from bacterial lysate to near homogeneity following a protocol developed for other trypanosome glycolytic proteins [10]. The purified protein had an approximate molecular mass of 45 kDa, similar to the 46.6 kDa mass predicted for the 429 amino acid primary sequence. This protein is more similar to *Hs*ENO2 than the ENO from *Naegleria fowleri* (*Nf*ENO, Table 1, Appendix A). *Tb*ENO enzyme displayed Michaelis–Menten kinetics when 2-PG concentrations were increased (Figure 1A) and had a pH optimum of pH 8.5. The enzyme activity was severely impacted at pH < 8 (Figure 1B).

The phosphonate-based *Hs*ENO2 inhibitors developed for treatment of *ENO-1* deleted glioblastoma were tested against recombinant *Tb*ENO as a first step toward using them as potential leads against African trypanosomes (Table 2). *Tb*ENO is 63% identical to *Hs*ENO2 and shares residues required for interactions with phosphonate inhibitors, including Arg372 (in *Hs*ENO2), required to form a salt bridge with the compound ^7^, leading us to speculate that the agents might prove useful here.

SF2312 is a phosphonate antibiotic originally isolated from the actinomycete *Micromonospora* that was found to have inhibitory activity against ENOs from a variety of sources [9]. The SF2312 analog (1-hydroxy-2-oxopyrrolidin-3-yl) phosphonic acid (deoxy-SF2312) was a potent *Tb*ENO inhibitor (IC_50_ value of 0.60 ± 0.23 µM), while benzyl-deoxy-SF2312 was not active against the enzyme at 10 µM.

HEX [(1-hydroxy-2-oxopiperidin-3-yl) phosphonic acid], which had improved specificity for ENO2 [7], was less potent than SF2312, with an IC_50_ value of 2.1 ± 1.1 µM, suggesting that the larger six-membered ring structure was less suitable for *Tb*ENO engagement (Figure 1C). Supporting this possibility, HEPTA, a seven-membered analog of HEX, was not active against *Tb*ENO (Table 2).

Modification of the phosphonates to generate prodrugs did not alter enzyme inhibition, with POMHEX inhibiting the protein (IC_50_ values of 2.8 ± 0.64 µM), at a level similar to the value seen with the unmodified inhibitor. This is distinct from what has been observed for an ENO from the pathogenic free-living amoebae *Naegleria fowleri*, *Nf*ENO, which was not inhibited by the POM derivatives, likely for steric reasons (Milanes et al., unpublished). Modification of the hydroxamates with benzyl moieties inactivated the inhibitors, supporting the likelihood that the hydroxamate portion of HEX is important for activity. This is consistent with the observation that benzylHEX was not active against ENO2 [7].

### 3.2. Molecular Modeling of HEX Compounds Bound to TbENO

Using the *Tb*ENO crystal structure bound to phosphoenolpyruvate (PEP) (Figure 2A,B) [5,6], we performed molecular docking simulations of the compounds with *Tb*ENO (PDB 2PTY) using Autodock Vina. Interestingly, HEX binds orthosterically to *Tb*ENO with a calculated binding affinity of −7.5 kcal/mol, and interacts with key residues Gln-164, Ser-40, Ala-39, Ser-373, Arg-372, Lys-343, and Lys-394, similar to active site-bound PEP (Figure 2B,C). POMHEX adopts a binding conformation like HEX, with the pivaloyloxymethyl (POM) group involved in key interactions with the amino acid residues Ser-40, Gln-164, Lys-243, and Lys-294 at the *Tb*ENO active site, yielding a calculated binding affinity of –6.6 kcal/mol (Figure 2D). While the ENO active site from diverse organisms like Archaea, Bacteria, and Eukarya tend to be conserved [17,18], POMHEX binding is not universal as the inhibitor showed very weak binding interactions with active site residues when docked with human enolase 2 (*Hs*ENO2). This divergence could be the result of interference caused by the POM-protecting group, which may disrupt necessary interactions between the inhibitor phosphonate group and the enzyme. The observed binding interaction from the docking model and the inhibitory effect of POMHEX on *Tb*ENO further corroborates the conformational variability and structural flexibility of the *Tb*ENO active site compared with other enolases, as previously reported [7].

Deoxy-SF2312 is also predicted to bind to the active site of *Tb*ENO with a calculated binding affinity of −6.8 kcal/mol. Analysis of the docking pose (Figure 2E) suggests that deoxy-SF2312 adopts a binding conformation similar to PEP. HEPTA’s calculated binding affinity is weakened (−4.8 kcal/mol) with the seven-membered ring and is not as well tolerated as the five and six-membered rings of deoxy-SF2312 and HEX in the active site.

### 3.3. The Phosphonohydroxamates Are Potent Anti-Trypanosomal Compounds

To assess the potential of the ENO2 phosphonate inhibitors as leads against *T. brucei*, the impact of the agents on parasites grown in culture was assessed. POMHEX was the lone ENO inhibitor active against parasites, suggesting that prodrug modification may be required for accessing the target in the parasite. BSF parasites had an EC_50_ value of 0.61 ± 0.08 µM while PF parasites cultured in standard (5 mM) glucose levels had an EC_50_ value of 0.81 ± 0.07 µM (Table 2 {Bauer, 2013 #3783}{Bauer, 2017 #5013}). PF parasites cultured in glucose-free medium were less sensitive to POMHEX, with an EC_50_ value of 5.0 ± 0.14 µM.

We hypothesized that cellular esterases may liberate HEX from POMHEX by acting on the pivaloyloxymethyl (POM) protecting group. The collapse of the POM group would liberate an equivalent of pivalate and formaldehyde along with unmodified HEX [19]. Thus, we sought to rule out the possibility that the mechanism of action of POMHEX could include cytotoxicity due to the release of formaldehyde. To interrogate this concern, we included formaldehyde with HEX in viability assays, which did not improve toxicity of the agent. HEX alone had an EC_50_ value of 10 ± 1.0 µM, while HEX with 2 molar equivalent formaldehyde had an EC_50_ value of 10 ± 1.4 µM, a notable difference from the EC_50_ value of POMHEX (0.61 ± 0.08 µM).

Incubation of BSF parasites that express glycosomally localized eYFP (BSF pXS6-eYFP-PTS1) with vehicle for 2.5 h had very little impact on cellular morphology. Incubation with POMHEX, however, led to a rapid change to overall cell shape (Appendix A). PF cellular and glycosomal morphology remained largely unchanged by a similar exposure to POMHEX. Interestingly, the abundance and distribution of the glycosome-resident protein aldolase in BSF was altered by POMHEX treatment (Appendix A). Aldolase levels were decreased 28% (with three other replicates revealing a 28-55% range of reduction) in treated cells when compared to vehicle-treated cells.

### 3.4. TbENO Inhibitor Toxicity Is Not Associated with Increased Cellular ROS Levels

Parasite cellular responses to toxins vary, but frequently anti-trypanosomal agents induce oxidative stress as part of their lethal effect [20,21]. To determine if the ENO2 inhibitors triggered oxidative stress, parasites were briefly exposed to high concentrations of POMHEX and then ROS levels were measured. Surprisingly, treatment with POMHEX (either 50 or 100 µM) led to a significant reduction in ROS levels when compared to untreated parasites, while treatment with hydrogen peroxide, a known precipitant of ROS in trypanosomes [22], caused an increase in reactive species (Figure 3A). To ensure that the reduction in ROS levels was not a consequence of measuring the levels in dead cells, parasite viability was assessed using PI exclusion (Figure 3B, Appendix A). Treatment with 50 µM POMHEX did not lead to increased cell death, unlike the higher concentration of the agent, indicating that the reduction in ROS production was not a result of increased cell death in the culture.

## 4. Discussion

The kinetoplastid parasites remain an important global cause of disease, with infection or the threat of infection impacting millions of people. One of the reasons these diseases are still prevalent is due to the limited number of efficacious treatment options. Here, we have found that phosphonohydroxamate ENO inhibitors are active against the African trypanosome ENO (*Tb*ENO), and are potent anti-trypanosomal compounds when tested against cultured pathogenic stage parasites.

The ENO2-specific inhibitor HEX has excellent pharmacodynamic and pharmacokinetic properties and is extremely well-tolerated in mammalian animal models, including non-human primates [7]. Further, HEX was shown to be curative for *ENO1*-deficient tumors in a rodent model, supporting brain distribution of the agent. This is a key requirement when considering the development drugs for treating African sleeping sickness, as parasites in the brain cause altered host sleep patterns (hence the name of the disease) and are ultimately responsible for host death. Therefore, compounds like HEX that can reach the brain are critical for resolution of infection.

We speculate that POMHEX may be toxic to BSF trypanosomes as a result of inhibiting glycolysis. The impact of the agent on aldolase levels was unanticipated. The causes and consequences of the reduction in aldolase protein are unclear but may reflect a cellular effort to prevent the inappropriate influx of the accumulated glycolytic intermediates that result from ENO inhibition into the gluconeogenic pathway. Alternatively, the accumulated intermediates may be perceived as a signal of sufficient glycolysis to meet the metabolic needs of the cell, leading to downregulation of aldolase levels.

POMHEX activity against PF parasites varied depending on the presence of glucose. Those cells cultured in the presence of the hexose, a condition in which sugar is known to be the preferred metabolite [23], were as sensitive to POMHEX as BSF parasites. However, PF parasites cultured in the absence of glucose, which are known to utilize other carbon sources, were less sensitive. This was likely because glycolysis was no longer serving as the predominant source of ATP. They remain modestly sensitive, however, possibly reflecting the role of *Tb*ENO in gluconeogenesis, an essential pathway [24].

Because ENOs in other systems play important non-glycolytic/non-gluconeogenic roles, it is possible that the toxicity in trypanosomes could also be in part due to other impacts. For example, ENOs in some bacterial systems are involved in the regulation of mRNA stability [25], while in mammals, one ENO isoform serves as a structural component of the eye lens [26]. These moonlighting functions raise the possibility that toxicity in the trypanosomes results from the impact of the agent on glucose metabolism and potentially other ENO functions.

Why is ROS production reduced in parasites treated with POMHEX? POMHEX may lead to the accumulation of metabolic intermediates upstream of ENO, including glucose-6-phosphate (G6P) [7,9]. G6P feeds into other critical pathways, including the pentose phosphate pathway, potentially resulting in the increased reduction in equivalent levels that would lead to reduced ROS.

In their current form, the ENO2 phosphonate inhibitors are not likely to be useful for treatment of trypanosome infections. The poor permeability resulting from the phosphonate group, an issue noted in the development of the agents as anti-cancer compounds [9], also likely impacts their activity against trypanosomes. Esterification with POM groups helped to resolve this, at least against cultured parasites grown in medium supplemented with heat-inactivated serum. However, the performance of POMHEX against cultured parasites is likely not indicative of its in vivo performance, as the POM groups are susceptible to serum esterases. So, the premature removal of these in the sera prior to uptake by the parasite would functionally inactivate the compounds. Nevertheless, the potent activity of the phosphonate ENO inhibitors serves as an excellent starting point for the development of new analogs for the treatment of kinetoplastid diseases.

## Figures and Tables

**Figure 1 pathogens-12-01290-f001:**
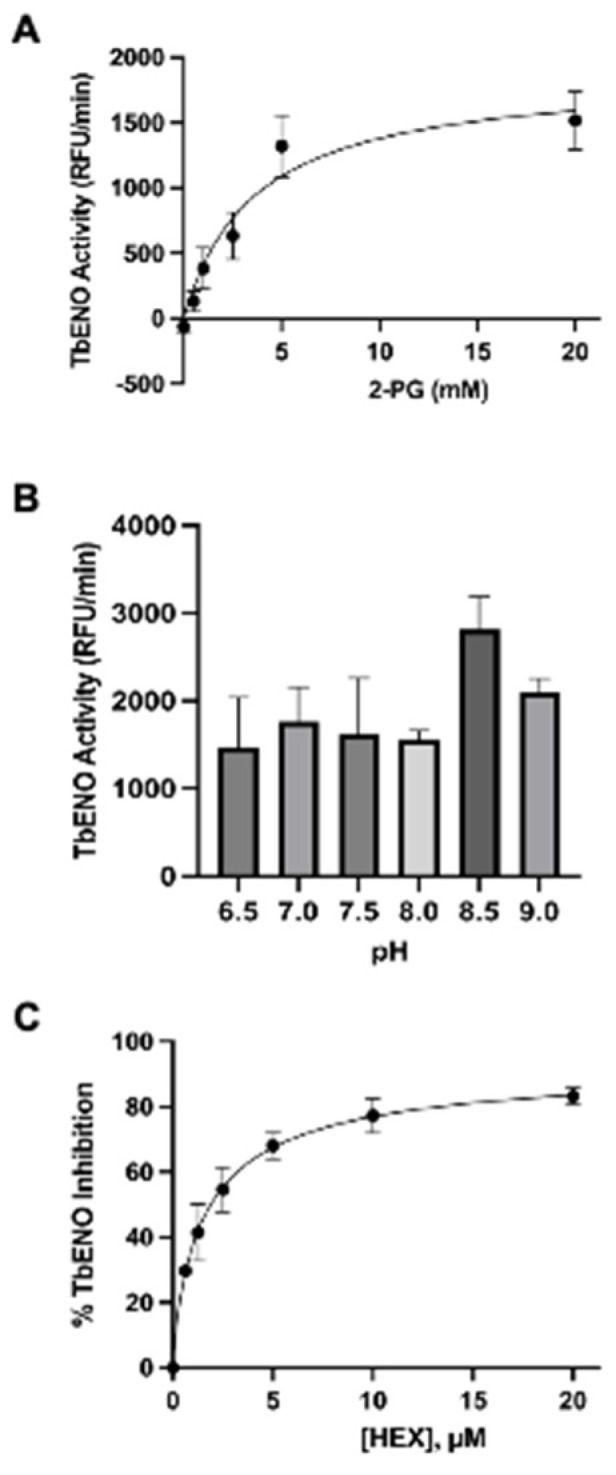
Effects of differing 2-PG concentrations and pH on *Tb*ENO activity. (**A**) *Tb*ENO (~50 nM) activity was measured in response to increasing 2-PG concentrations, from 0.5 to 20 mM. (**B**) *Tb*ENO (~50 nM) activity was measured at different pH values, from pH 6.5 to 9. Three different buffers were used for the assays: MES (pH 6.5), HEPES (pH 7-8.5), and sodium borate (pH 9.0). (**C**) *Tb*ENO (~50 nM) inhibition with different concentrations of HEX (0-20 µM). All assays were performed in triplicate and standard deviations are noted.

**Figure 2 pathogens-12-01290-f002:**
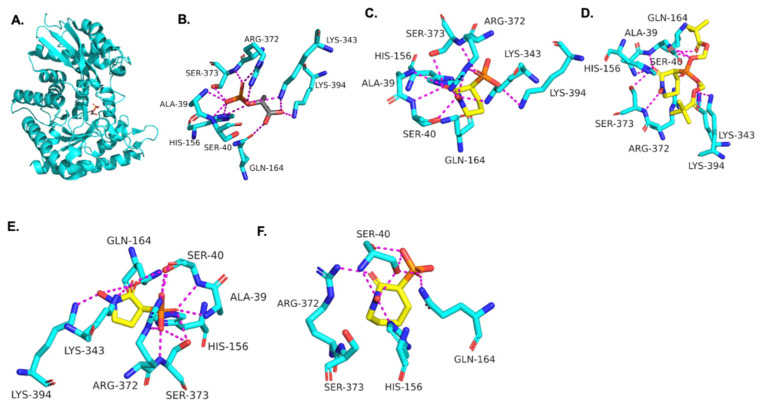
(**A**) Crystal structure of *Tb*ENO complexed with PEP (PDB 2PTY, 2.00 Å). Experimental binding poses of (**B**) PEP, (**C**) HEX, (**D**) POM-HEX, (**E**) deoxy-SF2312, and (**F**) HEPTA at the active site of *Tb*ENO, respectively.

**Figure 3 pathogens-12-01290-f003:**
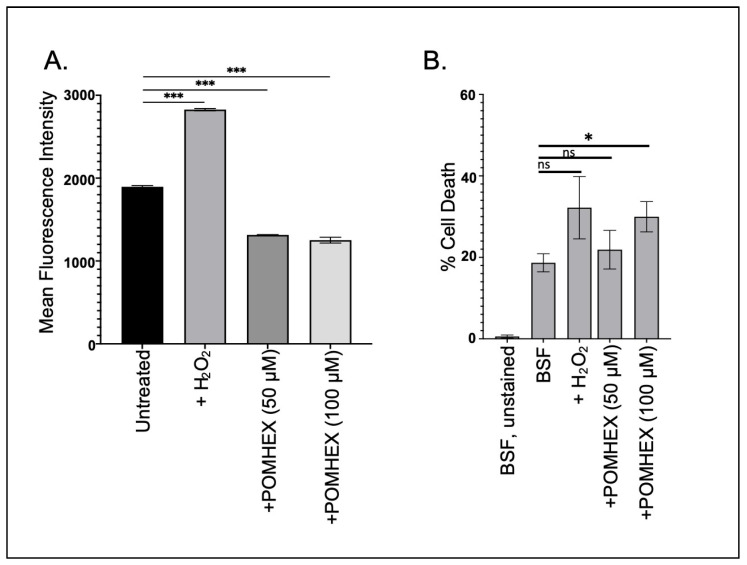
POMHEX leads to reduced ROS in BSF parasites. (**A**) Parasites (5 × 10^6^) were treated with either H_2_O_2_ (125 μM), POMHEX (50 or 100 μM), or equal volume DMSO, and analyzed by flow cytometry for ROS production for median fluorescence intensity (MFI) (ns: not statistically significant, *** *p* < 0.001). Data are representative of one technical triplicate and standard deviations are shown. (**B**) To monitor apoptotic-like responses, parasites were treated with H_2_O_2_, POMHEX, or DMSO and then stained with propidium iodide before analysis using cytometry. MFI was calculated and expressed as the percent of propidium iodide positive cells in the total population (ns: not statistically significant, * *p* < 0.05).

**Table 1 pathogens-12-01290-t001:** Comparison of the sequences of *Tb*ENO, *Nf*ENO, and human ENO proteins.

	*Tb*ENO
*Nf*ENO	46 (62)
*Hs*ENO2	63 (78)

Numbers are % identity and % positives. The sequences used for comparison were *Nf*ENO (NfTy_054390) and *Hs*ENO2 (*ENOG_Human*, P09104).

**Table 2 pathogens-12-01290-t002:** Phosphonate ENO2 inhibitors inhibit *Tb*ENO and have anti-trypanosomal activity.

Compound	Structure	*Tb*ENO IC_50_ (µM)	*Hs*ENO2 IC_50_ (µM) ^1^	*T. brucei* BSF EC_50_ (µM)
**HEX**	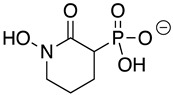	2.1 ± 1.1	0.36 ± 0.04	>10
**BenzylHEX ^2^**	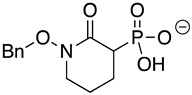	>10	>10	>10
**POMHEX**	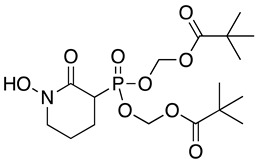	2.8 ± 0.64	>10	0.61 ± 0.08
**Deoxy-SF2312**	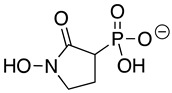	0.60 ± 0.23	0.10 ± 0.04	>10
**Benzyl-deoxy-SF2312**	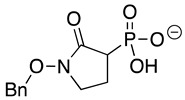	>10	>10	>10
**HEPTA**	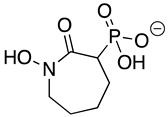	>10	5.8 ± 0.88	>10
**Benzyl-HEPTA**	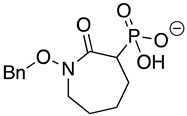	>10	>10	>10

^1^ Included for comparative purposes. ^2^ Bn, -CH_2_-C_6_H_5._

## Data Availability

Not applicable.

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
