# Peer review of "Enolase Inhibitors as Early Lead Therapeutics against Trypanosoma brucei"

_pathogens, 2023, doi:10.3390/pathogens12111290_

Round 1
Reviewer 1 Report
Comments and Suggestions for Authors
Page 5 Lines 251-253: The authors refer to the role that aldehydes may or not have in the potency of POMHEX in the viability of BSF parasites and the addition formaldehyde cultures, but there is no reference to this in the methods unless implied in the reference therein. Additionally, none of the figures the or results show where this is done. Please clarify, remove or provide better information.
Page 10 Lines 345-346: The cause of death to BSF parasites by POMHEX is not obvious as the EC50 is about 4.5x lower than the IC50 for the enzyme. These differences in the EC50 vs IC50 concentrations could also mean that POMHEX kills the BSM parasite by a different mechanism or that it gets concentrated in the glycosomes. It could be more accurate to state the hypothetical or most likely mechanism by which that POMHEX kills the BSF parasites.
Page 8 Lines 284-288: No description for Figure 1C in the legend.
Author Response
Enolase inhibitors as early lead therapeutics against Trypanosoma brucei
Roster et al.
Manuscript ID: pathogens-2627041
We appreciate the enthusiasm that the reviewers had for our manuscript and have addressed their concerns below. Changes to the text have been highlighted to aid in finding them for rereview.
Comments to the Author
Reviewer 1
- “Page 5 Lines 251-253: The authors refer to the role that aldehydes ….clarify, remove or provide better information”
We have addressed this issue by inclusion of additional details about the experiment and a reference that describes the likely results of the metabolism of the pro-drug.
- “Page 10 Lines 345-346: The cause of death to BSF parasites by POMHEX …could be more accurate to state the hypothetical or most likely mechanism by which that POMHEX kills the BSF parasites.
We have modified the text to indicate as suggested .
- “Page 8 Lines 284-288: No description for Figure 1C in the legend.”
We have modified the figure legend to include information about Figure 1C .
Reviewer 2 Report
Comments and Suggestions for Authors
The manuscript by Roster et al., showing Enolase inhibitors as early lead therapeutics against Trypano-2 soma brucei is overall an interesting work. However the data presented in the manuscript needs clarity.
1. The authors should upload the sequence files as supplementary.
2. For figure 3A, flow plots with gating startegy should be shown in supplementary.
3. Fig 3B. The authors show their flow gating for Annexin and PI staining gating strategy in supplementary. Did they use double positive cells for qunatification of cell death?
4. For figure S1A, the image shows only 1 parasite, can the suthors include a field that shows multiple?
With this said I believe that the manuscript is only descriptive and does not advance much our knowledge about using current Enolase inhibitors for treatment of Trypano-2 soma brucei.
Comments on the Quality of English Language
Language need moderated editing
Discussion can be rephrased
Author Response
Enolase inhibitors as early lead therapeutics against Trypanosoma brucei
Roster et al.
Manuscript ID: pathogens-2627041
We appreciate the enthusiasm that the reviewers had for our manuscript and have addressed their concerns below. Changes to the text have been highlighted to aid in finding them for rereview.
Reviewer 2
- “The authors should upload the sequence files as supplementary.“
We have included the sequence files as an alignment in Supplementary Figure S1.
- “For figure 3A, flow plots with gating … in supplementary.”
The data was not gated for this figure. Rather, the cytometer output was used to generate histograms (PC5.5- H on the X axis) and the associated software (FCSexpress) was used to calculate the mean fluorescent intensity (MFI) for the total population. As we piloted this experiment, we did establish a gate around the denser area of the untreated control population to determine if outliers were unduly influencing the output. This did not alter the results for the treated samples, so gates were not used. To collect the data for this experiment with three biological replicates, there were a total of 108 plots generated.
- “Fig 3B. The authors show their flow gating …of cell death?
The ROS measurements and the PI exclusion experiments were separate experiments. Figure 3B was generated by PI staining alone. We have included a representative example in a new Supplemental Figure (Supplemental Figure 3). Gates were established to capture most of the unstained cell population. The PI + H2O2 sample was then scored to confirm that the new population that appeared was fully in the PI-positive gate. These parameters were then applied to the rest of the samples.
- “For figure S1A, the image shows …shows multiple?
We have now included an image showing multiple cells.
Reviewer 3 Report
Comments and Suggestions for Authors This manuscript presents research on the potential use of phosphonohydroxamate ENO inhibitors as anti-trypanosomal compounds. While the study offers some interesting insights, there are several shortcomings and areas for improvement that need to be addressed. Mazar comments: 1. The introduction is somewhat unclear and lacks a comprehensive overview of the African sleeping sickness issue. It does not effectively set the context for the reader and lacks a clear hypothesis or research question. 2. Methods section is insufficiently detailed. There is a lack of information regarding the sources of reagents, controls, and sample sizes. The precise methods for molecular modeling and docking are not explained adequately. 3. The presentation of results is incomplete and lacks necessary statistical analyses(Page 8 line301-302). At least 3 biological replicates should be needed. 4. Authors should generate some mutants’ proteins on the basis molecular docking and perform in-vitro and in-vivo experiments. 5. Some content is repetitive, with information presented in the introduction being reiterated in the results section. This redundancy should be avoided to maintain clarity. In summary, while the research described in the manuscript is potentially significant but it’s in preliminary stage and there are multiple areas that require improvement, including more detailed methods, additional experimental data, and a more thorough discussion of the implications of the findings.
Author Response
Enolase inhibitors as early lead therapeutics against Trypanosoma brucei
Roster et al.
Manuscript ID: pathogens-2627041
We appreciate the enthusiasm that the reviewers had for our manuscript and have addressed their concerns below. Changes to the text have been highlighted to aid in finding them for rereview.
Reviewer 3
- “The introduction is … hypothesis or research question.”
We have clarified the issue with trypanosomes and human health and have modified our introduction to include an explicit hypothesis of the work.
- “Methods section is insufficiently …docking are not explained adequately.”
We have expanded the methods for docking and modeling, per the request.
- “The presentation of results … should be needed.”
We have clarified replicate number throughout the manuscript.
- “Authors should generate some …vivo experiments.”
This experiment is unfortunately not feasible currently as it is beyond the capacity of the current team and budget. We hope to pursue this in the future.
- “Some content is …. avoided to maintain clarity”
We have edited the discussion to limit the repetition of ideas related to HEX background, which were already found in the introduction.
Round 2
Reviewer 2 Report
Comments and Suggestions for Authors
The authors have made the necessary changes as per reviewers comments.
Comments on the Quality of English Language
Minor editing of the language needed
Reviewer 3 Report
Comments and Suggestions for Authors
The authors fulfilled the suggestions and I did not have any other comments.